# Transcriptome Sequencing Reveals Pathways Related to Proliferation and Differentiation of Shitou Goose Myoblasts

**DOI:** 10.3390/ani12212956

**Published:** 2022-10-27

**Authors:** Jiahui Chen, Shuai Zhang, Genghua Chen, Xianqi Deng, Danlu Zhang, Huaqiang Wen, Yunqian Yin, Zetong Lin, Xiquan Zhang, Wen Luo

**Affiliations:** 1Department of Animal Genetics, Breeding and Reproduction, College of Animal Science, South China Agricultural University, Guangzhou 510642, China; 2Guangdong Provincial Key Laboratory of Agro-Animal Genomics and Molecular Breeding, and Key Laboratory of Chicken Genetics, Breeding and Reproduction, Ministry of Agriculture and Rural Affair, South China Agricultural University, Guangzhou 510642, China

**Keywords:** Shitou goose, RNA-seq, mRNA, long non-coding RNAs, alternative splicing

## Abstract

**Simple Summary:**

The Shitou goose, the largest cultivated goose breed in China, has high research value in meat traits. Muscle development is regulated by genes related to myoblast proliferation and differentiation. In this study, the mRNA and lncRNA expression profiles of Shitou goose myoblast in proliferation and differentiation were constructed by Illumina sequencing. A total of 1664 differentially expressed (DE) mRNAs and 244 DE-lncRNAs were identified between the two periods. Functional annotation showed that the DE-mRNAs and DE-lncRNAs were mainly enriched in the Wnt signaling pathway. These results provide new insights into the mechanism of muscle growth and development in large goose breeds.

**Abstract:**

Chinese Shitou goose is a type of large goose with high meat yield. Understanding the genetic regulation of muscle development in Shitou goose would be beneficial to improve the meat production traits of geese. Muscle development is regulated by genes related to myoblast proliferation and differentiation. In this study, the RNA-seq method was used to construct the mRNA and lncRNA expression profiles of Shitou goose myoblasts and myotubes. A total of 1664 differentially expressed (DE) mRNAs and 244 DE-lncRNAs were identified. The alternative mRNA splicing in proliferation and differentiation stages was also analyzed. Notably, pathways enriched in DE-mRNAs, DE-splicing transcripts, and DE-lncRNAs all point to the Wnt signaling pathway, indicating that the Wnt signaling is a key regulatory pathway of muscle development in Shitou goose. We also constructed the interactive network of DE-lncRNAs and DE-mRNAs and revealed some key genes of lncRNAs regulating the proliferation and differentiation of myoblasts. These results provide new insights for the study of the muscle development of the Shitou goose.

## 1. Introduction

In animal husbandry, meat is the main product [1], and most of the selection and breeding of meat-raised animals is focused on improving the percentage of retail cuts of the animals. As the main component of the animal body, skeletal muscle accounts for about 35% of body weight [2], and the development and growth of muscle is the key factor to provide enough meat for humans. Understanding the growth and development of skeletal muscle is important to improve the percentage of retail cuts of livestock. The growth and development of skeletal muscle is an extremely complex process, including the directed differentiation of progenitor cells, the proliferation and differentiation of myoblasts, the fusion of myocytes, and finally the formation of multinucleated muscle fibers with contractile function [3]. There have been many reports on the muscle development of meat-raised animals, especially pigs, chickens, cattle, and goats [4], but the research on the muscle development of geese needs to be increased.

Shitou goose is a large and excellent breed of goose native to southern China. It is currently raised and eaten in more than 20 provinces in China [5]. China is a big consumer of goose meat and is the largest meat goose breeding country. In 2020, about 740 million meat geese were sold worldwide, an increase of about 17.5% compared with 2019 [6]. Goose meat is a popular green food with low fat, low cholesterol, and high protein. As the largest goose species in China, the Shitou goose is famous for its fast-growing, early sexual maturity, rich muscles, and excellent roughage resistance. The muscle growth performance of the goose is closely related to meat production, which is linked to the supply of goose meat and economic benefits to the farmer. Shitou geese are suitable for the study of goose muscle growth and development due to the advantages of fast growth and high meat production. A total of 290 differentially expressed genes (DEGs) were identified between large-size Shitou goose and small-size Wuzong goose by using RNA-seq [7]. An interaction network analysis showed that the genes related to protein synthesis were upregulated in expression in the Shitou goose compared to the Wuzong goose [7]. Histological analysis revealed that the Shitou goose had higher myofiber density and larger myofiber diameter at embryonic stages than the Wuzong goose [8]. Several candidate myo-genes, such as *MYH1*, *MYH2*, and *MYH7*, were identified that may contribute to the distinct muscle development processes between the two goose breeds during embryonic stages [8]. However, the key genes and pathways responsible for high meat production in Shitou goose remain highly unclear.

The splicing of precursor messenger RNA (Pre-mRNA) occurs in most eukaryotes as a process of removing noncoding regions (introns) and joining coding regions (exons). The splicing of Pre-mRNA is accomplished by the sequential assembly of five small nuclear RNA-proteins (snRNPs) called spliceosomes [9,10]. Alternative splicing is considered a key factor in the increased cellular and functional diversity and complexity of higher eukaryotes [11,12]. Variability in splicing patterns is a major source of protein diversity in the genome [13]. In humans, two-thirds of genes contain one or more alternatively spliced exons [14]. However, the alternative splicing diversity of goose mRNA has never been studied before, especially during muscle development. The functional diversity of myogenic genes caused by alternative splicing may be a key factor affecting muscle development in the goose.

Long non-coding RNAs (lncRNAs) refer to RNA fragments longer than 200 nucleotides [15]. In recent years, more and more attention has been paid to the regulatory role of non-coding RNA in muscle development. It has been pointed out that lncRNA plays an important regulatory role in different stages of myogenesis [16,17]. For example, lncRNA H19 promotes muscle differentiation and skeletal muscle regeneration [16], and lncMYOD inhibits myoblast differentiation and increases fast muscle fibers but reduces slow muscle fibers [18]. The role of lncRNA is generally to regulate transcriptional and post-transcriptional processes by interacting with transcription factors and mRNA [19]. In this study, we will focus on the interaction between lncRNA and mRNA during myogenesis in the goose. By using RNA-seq, we analyzed the expression difference in lncRNA and mRNA between the proliferation and differentiation stages of myoblast derived from Shitou geese. An interactive network between DE-lncRNAs and DE-mRNAs was constructed to identify some key genes and lncRNAs involved in goose muscle development. We think this study will provide new insights into the genetic regulation mechanism of fast-growing meat geese.

## 2. Materials and Methods

### 2.1. Isolation and Culture of Goose Myoblasts

Ten embryos of Shitou geese, hatched for 15 days, were used for isolation of a primary myoblast. After knocking open the air chamber end with surgical tweezers, we placed the goose embryo into a large culture dish with a diameter of 10cm, then amputated the legs and removed the bones, skin, and fascia. The leg muscles were minced with curved scissors in the culture medium and then digested with trypsin at 37 °C for 20–30 min. A DMEM medium containing 20% fetal bovine serum (FBS) (Gibco, Grand Island, USA) was added to terminate the digestion. The mixed medium was centrifuged at 1200 g for 5 min, and then the supernatant was discarded. A DMEM medium with 15% FBS was added to resuspend the cells and then put into a 10 cm diameter petri dish. Serial plating was performed to enrich myoblasts and eliminate fibroblasts. We divided the mixed primary myoblasts into 6 cultures. Three cultures were used to collect proliferating myoblasts (GM), and the other three cultures were induced into differentiation and then used to collect differentiated myotubes (DM). The differentiation of myoblasts was induced by replacing 20% FBS with 2% horse serum (HS) (Biosharp, Hefei, China).

### 2.2. RNA Extraction, cDNA Synthesis, Fluorescence Quantitative PCR

Total RNA was extracted from goose myoblasts using MagZol Reagent (Magen, Guangzhou, China). The Evo M-MLV RT Kit (Agbio, Guangzhou, China) was used for reverse transcription to synthesize cDNA. MonAmp™ ChemoHS qPCR Mix (Monad, Guangzhou, China) was used to perform qRT-PCR. All reactions were set to three replicates. The 2^−∆∆Ct^ method was used to measure gene expression with GAPDH as the reference gene [20,21].

### 2.3. Library Construction and Illumine Sequencing

The proliferative myoblasts of Shitou geese were cultured to 70% density in the petri dish, then washed with PBS three times, and directly treated with Trizol reagent (Invitrogen, Carlsbad, CA, USA). The differentiated myoblast needs to be cultured with a DMEM of 2% horse serum when the density reaches 80–90%. After three days of differentiation, the samples were collected with Trizol. Total RNA was isolated and purified using Trizol reagent following the manufacturer’s procedure. The RNA amount and purity of each sample were quantified using NanoDrop ND-1000 (NanoDrop, Wilmington, DE, USA). The RNA integrity was assessed by Agilent 2100 with RIN number >7.0. Approximately 5 ug of total RNA was used to deplete ribosomal RNA according to the manuscript of the Ribo-Zero™ rRNA Removal Kit (Illumina, San Diego, CA, USA). Paired-end sequencing was performed on an Illumina Novaseq™ 6000 (LC Bio, Hangzhou, China) following the vendor’s recommended protocol.

### 2.4. Bioinformatics Analysis

RNA-seq data from six geese have been deposited in NCBI Gene Expression Omnibus (GEO) data repository under accession number GSE213147. Cutadapt (https://cutadapt.readthedocs.io/en/stable/, accessed on 3 December 2020) and in-house perl scripts were used to remove reads containing undetermined bases, low-quality bases, or those that contained adaptor contamination. FastQC (http://www.bioinformatics.babraham.ac.uk/projects/fastqc/, accessed on 14 December 2020) was used for quality control. Bowtie2 [22] was used to map reads to the genome of *Anser cygnoides* (assembly GooseV1.0 (http://asia.ensembl.org/Anser_cygnoides/Info/Index/, accessed on 26 December 2020)). The mapped reads of each sample were assembled using StringTie [23]. Then, all transcriptomes from Shitou goose were merged to reconstruct a comprehensive transcriptome using perlscript and gffcompare. After the final transcriptome was generated, StringTie and Ballgown [24] were used to estimate the expression levels of all transcripts. Differential expression analysis was performed using the edgeR according to the edgeR User’s Guide [25]. The differentially expressed mRNAs and lncRNAs were selected with log2 (fold change) > 1 or log2 (fold change) < −1 and with statistical significance (adjusted *p*-value < 0.05) by R package edgeR. See the Appendix A for the RNA sequence data processing flow chart (Appendix A).

The lncRNA prediction software CPC (coding potential calculator) [26] and CNCI (coding non-coding index) [27] were used to predict the lncRNA transcripts larger than 200 bp. All transcripts with CPC score < −1 and CNCI score < 0 were removed. The codes for Gene differential splicing analysis can be referred to Appendix A.

The alternative splicing transcripts in RNA-seq data were analyzed by rMATS [28]. Differential splicing genes were selected with log2 (fold change) > 1 and abs(lncLeveldiff) > 0.1. Gene ontology (GO) and Kyoto Encyclopedia of Genes and Genomes (KEGG) analysis of the enriched genes was performed using the Database for Annotation, Visualization and Integrated Discovery (DAVID) (https://david.ncifcrf.gov/tools.jsp/, accessed on 22 March 2021) and R Studio. Genes expressed in our samples were used as the background. The R Studio code can refer to Appendix A.

### 2.5. Construction of DE-lncRNAs-DEGs Interaction Network

We searched for all of the DEGs 100 kb upstream and downstream of differentially expressed lncRNA by the UCSC Genome Browser database [29]. These genes that are genomically adjacent and coexpressed in the expression pattern are likely to be the cis-target genes of the lncRNA. The lncRNA-gene Interaction was constructed by Cytoscape software (https://cytoscape.org/, accessed on 14 June 2021). Functional analysis of the cis-target genes was conducted by BLAST2GO [30]. Differences at *p* < 0.05 were regarded as statistically significant.

### 2.6. Statistical Analysis of qPCR Results

For the qPCR experiment, each sample was analyzed based on results that were repeated in triplicate and analyzed by Graphpad 8.0 (GraphPad Software, San Diego, CA, USA). The statistical significance of differences between the two groups was determined by a standard Student’s *t*-test. In all cases, differences at *p* < 0.05 were regarded as statistically significant (Significance is coded as * for *p* < 0.05 and ** for *p* < 0.01).

## 3. Results

### 3.1. Isolation and Transcriptome Sequencing of Shitou Goose Myoblasts

To identify the genes involved in the development of Shitou goose muscle during the proliferation and differentiation of myoblasts, we used the Illumina HiSeq platform to sequence the transcriptome of cells in the proliferation (GM) and differentiation (DM) stages (Figure 1A). The box diagram (Figure 1B) showed that the sequencing quality of the six sequencing samples is relatively consistent. The percentage of unique Mapped reads of the samples is greater than 70%, the Q30 value is greater than 98.4%, and the GC base content is between 47–48% (Table 1). There were 1664 differentially expressed genes (DEGs) between the proliferation and differentiation stages of Shitou goose myoblasts, including 685 up-regulated DEGs and 845 down-regulated DEGs (Figure 1C). The top ten differentially expressed genes are listed in Table 2.

### 3.2. Functional Analysis of DEGs between Shitou Goose Myoblasts and Myotubes

We then performed GO and KEGG analysis on the DEGs to identify important genes and pathways during goose muscle development. GO functional enrichment analysis found that down-regulated DEGs were significantly enriched in muscle development, such as positive regulation of cell proliferation, positive regulation of cytosolic calcium, cell adhesion molecule binding, and actin binding (Figure 2A). The up-regulated DEGs were significantly enriched in cellular calcium ion homeostasis, skeletal muscle tissue development, and calcium ion transmembrane transport (Figure 2B). KEGG pathway enrichment analysis found that the DEGs were significantly enriched in the pathways related to cell growth and metabolism. The down-regulated DEGs enriched pathways include the PI3K-Akt signaling pathway, the JAK-STAT signaling pathway, and the ECM-receptor interaction (Figure 2C). The up-regulated DEGs enriched pathways include the calcium signaling pathway, cardiac muscle contraction, and hypertrophic cardiomyopathy (Figure 2D). 

Next, Gene set enrichment analysis (GSEA) of the RNA-seq data demonstrated that the differentiation of goose myoblast led to the positive enrichment of muscle cell differentiation, muscle organ development, and cell adhesion (Figure 3A). Genes related to cell cycle and DNA duplication were enriched in proliferative cells (Figure 3B). We randomly selected some DEGs for qPCR verification. Muscle differentiation-related genes such as *MYH15* and *MyoM* are highly expressed in the differentiation stage. Cell cycle-related genes, such as *CDKN1A* and *CCNB1*, are significant differences in expression between the two stages (Figure 3C). These results are consistent with the RNA sequencing results (Figure 3D). 

### 3.3. Differential Splicing Analysis of mRNA during Goose Myoblast Differentiation

When DNA is transcribed into RNA, it is first transcribed into Pre-mRNA by RNA polymerase. Pre-mRNA is spliced into mature mRNA by the spliceosome, and the mature RNA can be further translated into protein [31,32]. Different types of cells can produce different splicing variants through variable splicing [33,34]. Therefore, we next analyzed the differentially splicing genes (DSGs) between goose myoblast and myotube (Table 3). A total of 854 DSGs were found between the two groups (Appendix A), and the main splicing type is skipped exon (SE), which accounts for 67.70% (Figure 4A). GO and KEGG analyses were performed on the 854 DSGs. A KEGG pathway analysis found that DSGs were mainly enriched in the Wnt signaling pathway and fatty acid biosynthesis (Figure 4B). A GO enrichment analysis found that the DSGs were enriched in skeletal muscle tissue development, sequence-specific DNA binding, and regulation of alternative mRNA splicing, via spliceosome (Figure 4C). We also analyzed genes that were both differentially expressed and differentially spliced. A total of 43 genes were significantly differentially expressed and differentially spliced between the two groups (Figure 4D, Appendix A). The functional analysis found that the 43 genes were significantly enriched in an extracellular matrix and actin binding (Table 4).

### 3.4. Differential Expression Analysis of LncRNA during Goose Myoblasts Differentiation

The development of myogenesis is regulated by epigenetic factors, including DNA methylation, RNA methylation, histone modification, and noncoding RNA [35,36,37]. There have been many reports on the research of lncRNA in poultry [38,39]. In this study, we examined the differential expression analysis of lncRNA in the proliferation stage and differentiation stage of Shitou goose myoblasts. A total of 244 DE-lncRNAs were found between the two groups (Figure 5A). A KEGG enrichment analysis found that DE-lncRNAs were enriched in the cGMP-PKG signaling pathway, the calcium signaling pathway, the Wnt signaling pathway, and the cAMP signaling pathway (Figure 5B). These pathways are related to muscle development [40,41]. A GO analysis of DE-lncRNA found that the functions of lncRNAs were mainly enriched in epigenetic regulation, such as transcriptional regulation and transcriptional regulation of an RNA polymerase promoter (Figure 5C). Next, we constructed a putative DE-lncRNA-DEGs interactive network involved in goose myogenesis. Among them, 316 lncRNAs were found to target 150 cis-DEGs, with a total of 631 pairs of lncRNA-mRNA connections (Figure 6, Appendix A). The expression relationship of 378 pairs of lncRNA-mRNA is positive, and the expression relationship of 253 pairs of lncRNA-mRNA is negative.

## 4. Discussion

In this study, we identified the genes and pathways related to the proliferation and differentiation of Shitou goose myoblasts through transcriptomics and spliceomics analysis. Since the proliferation and differentiation of myoblasts affect muscle growth and hypertrophy [42], the determination of related genes and pathways has important reference value for subsequent meat goose breeding. We also found that there was different splicing of multiple genes during myoblast proliferation and differentiation, indicating that one gene may play different roles in the regulation of muscle development.

Animal meat production mainly depends on the increase in the number of muscle fibers and muscle fiber hypertrophy. Previous studies have shown that the number of muscle fibers does not increase after birth and acquired muscle growth mainly depends on muscle fiber hypertrophy [43]. As a large goose species, the Shitou goose has faster muscle development before hatching, and it has more and larger muscle fibers than the small goose species [8]. Similarly, a chicken with high weight selection grows faster in the later stage of embryonic development than a chicken with low weight selection [44]. These results suggest that more and faster muscle fiber development before hatching is critical for the postnatal growth of the Shitou goose. The sequencing samples in this study are myoblasts isolated from 15 embryonic Shitou goose embryos. The myoblasts obtained in this period have active proliferation and differentiation potential [45], better reflecting the proliferation of myoblasts in the embryonic stage of Shitou goose and showing the process of differentiation and fusion of myoblasts to form polynuclear muscle fibers.

In recent years, it has become more and more popular to use bioinformatics to analyze changes in mRNA expression profiles of growth and development in non-model animals. As an excellent meat goose variety, it is very important to study the mRNA expression profile for the regulation of Shitou goose muscle development. In birds, muscle growth and development are regulated by a variety of signaling pathways, including the JAK-STAT signaling pathway [46], the MAPK signaling pathway [47,48], the PI3k pathway [49], the focal adhesion pathway [50], the TGF-β signaling pathway [51] and so on. In this study, we found that the JAK-STAT signaling pathway and PI3k pathway were significantly enriched in DEGs during goose proliferation (down-DEGs). Many studies on these signaling pathways related to muscle have been reported. For example, the JAK-STAT signaling pathway is related to satellite cell proliferation and muscle atrophy [52]. In the mouse model, the JAK-STAT signaling pathway is related to Critical Illness Myopathy (CIM), and Chronic JAK/STAT activation promoters loss of muscle mass and function [53]. The P13K-Akt signaling pathway at the GM stage is significantly enriched (Figure 2C), indicating that it plays an important role in the proliferation of Shitou goose myoblasts and skeletal muscle development. The PI3K Akt signaling pathway is highly conserved in animals [54] and plays an important role in normal cell activities related to cell growth, proliferation, movement, survival, metabolism, and apoptosis [55]. The PI3K-Akt signaling pathway is also associated with muscle-related diseases and muscle development [56], indicating that this pathway may play an important regulatory role in Shitou goose muscle development.

This study not only analyzed the DEGs based on transcriptome sequencing data but also analyzed the DSGs during goose myoblast differentiation. Splicing depends on the dynamic assembly of ribonucleoprotein complexes, including spliceosomes and a large number of helper proteins. The spliceosomes influence the fate of RNA and cell developmental processes by selecting alternative splicing sites [9]. At the single-cell level, the dynamic splicing of RNA is the key to the bioinformatics evaluation of cell fate [57]. This suggests that differential splicing may be an important factor for the smooth progress of cell differentiation during the muscle development of the Shitou goose. Some pieces of literature have pointed out that some signaling pathways can indirectly affect cell fate by affecting splicing factors, such as the MAPK signaling pathway [58] and the Wnt signaling pathway [59]. The data of this study also showed that the DEGs and DSGs during Shitou goose myoblasts differentiation were significantly enriched in the Wnt signaling pathway, indicating that this pathway may regulate the proliferation and differentiation of myoblasts by affecting gene signaling transduction and mRNA splicing.

## 5. Conclusions

In conclusion, we constructed the mRNA and lncRNA expression profiles of Shitou geese during myoblast differentiation. A total of 1664 DEGs, 854 DSGs, and 244 DE-lncRNAs were identified between myoblasts and myotubes. By analyzing the biological functions of DEGs, DSGs, and DE-lncRNAs, a set of signaling pathways, such as the Wnt signaling pathway, was found to be highly related to the muscle development of Shitou geese. The results of this study can provide a basis for a follow-up study on the mechanism of goose muscle growth and development.

## Figures and Tables

**Figure 1 animals-12-02956-f001:**
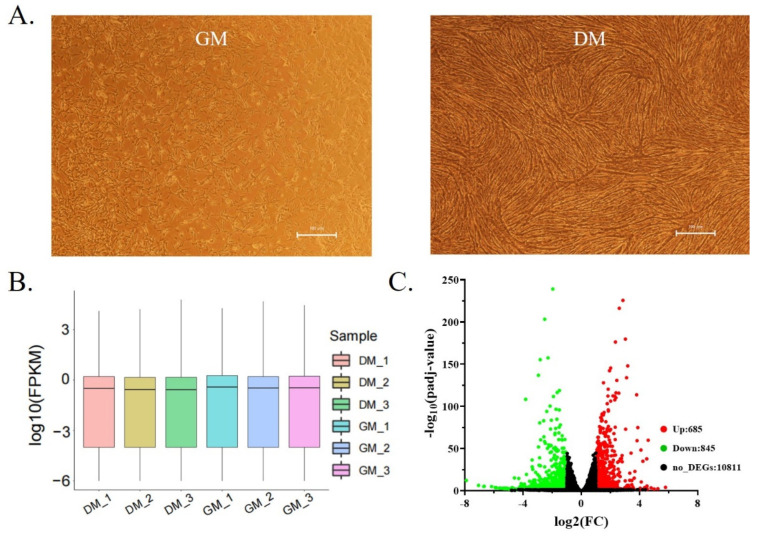
Shitou goose myoblasts isolation and transcriptome sequencing. (**A**) The state of proliferation and differentiation of Shitou goose myoblasts; (**B**) box diagram of transcriptome expression of each sample; (**C**) the volcano plot showing DEGs between GM and DM.

**Figure 2 animals-12-02956-f002:**
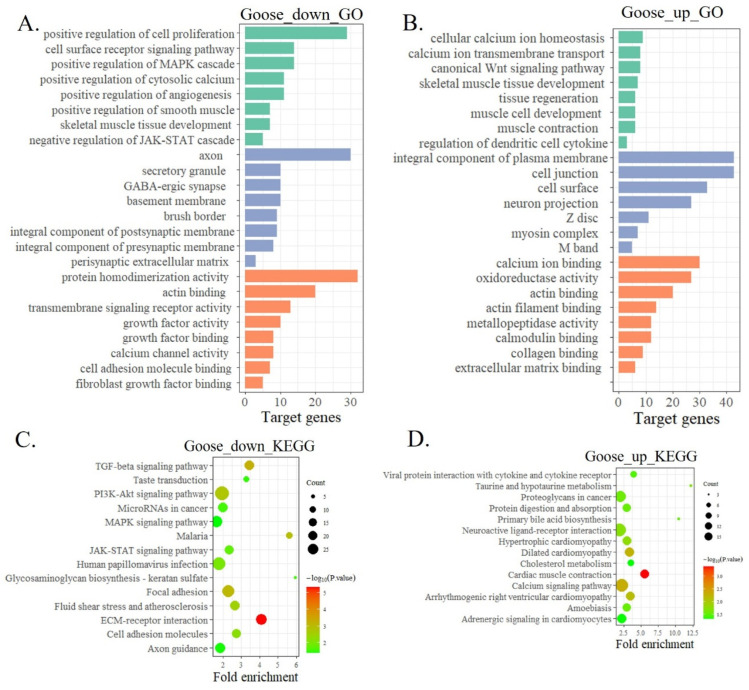
A functional analysis of DEGs between Shitou goose myoblasts and myotubes. (**A**) Gene Ontology enrichment of down-regulated DEGs; (**B**) gene ontology enrichment of up-regulated DEGs; (**C**) the significantly enriched KEGG pathways of down-regulated DEGs; (**D**) the significantly enriched KEGG pathways of up-regulated DEGs.

**Figure 3 animals-12-02956-f003:**
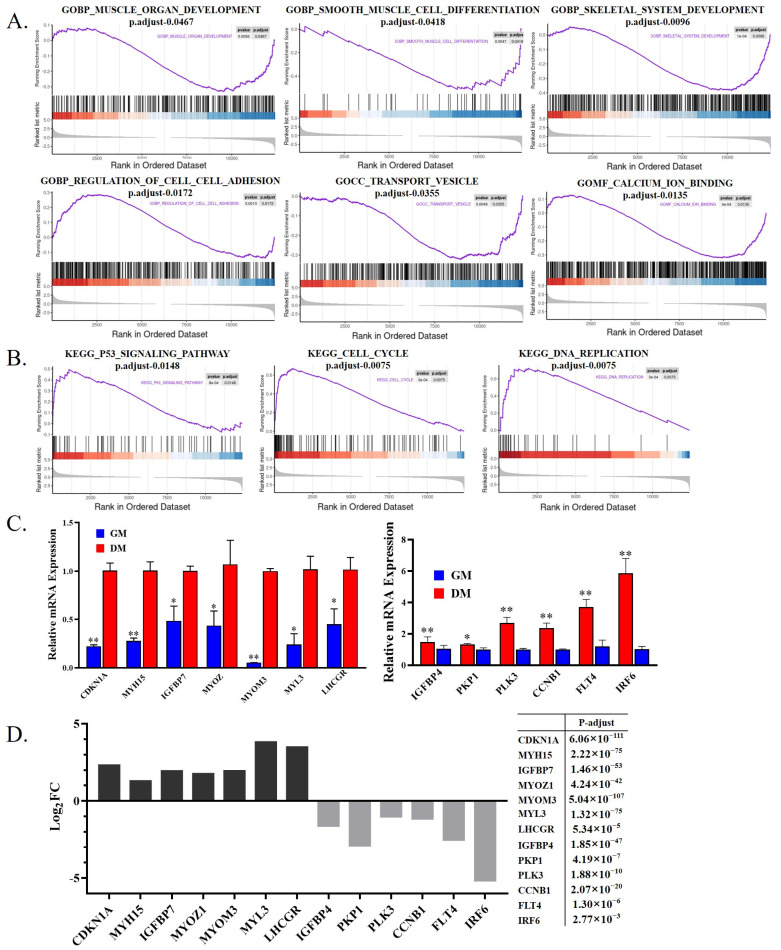
A gene set enrichment analysis of DEGs between Shitou goose myoblasts and myotubes. (**A**) Significantly enriched Gene Ontology terms by GSEA analysis; (**B**) significantly enriched KEGG pathways by GSEA analysis; (**C**) qPCR validation of some DEGs involved in muscle development; (**D**) RNA-seq results of some DEGs involved in muscle development. The data in C are mean ± S.E.M. with three replicates per group. One-sample *t*-test was used to assess the difference between the two groups. * *p* < 0.05; ** *p* < 0.01.

**Figure 4 animals-12-02956-f004:**
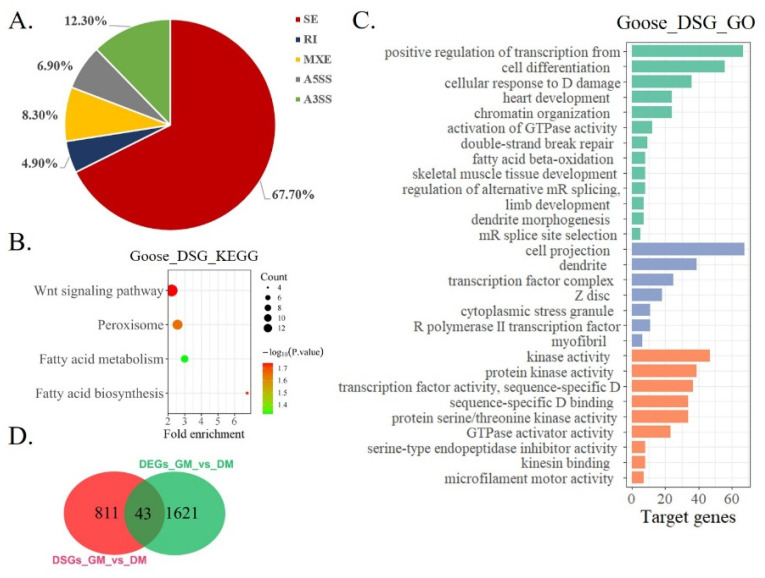
A differential splicing analysis of mRNA during goose myoblast differentiation. (**A**) The proportion of various types of DSGs. SE: Skipped Exon. RI: Retained Intron. MXE: Mutually Exclusive Exons. A5SS: Alternative 5′ Splice Site. A3SS: Alternative 3′ Splice Site; (**B**) KEGG enrichment analysis of DSGs between proliferative and differentiated myoblasts; (**C**) GO enrichment analysis of DSGs; (**D**) Venn diagram of DEGs & DSGs.

**Figure 5 animals-12-02956-f005:**
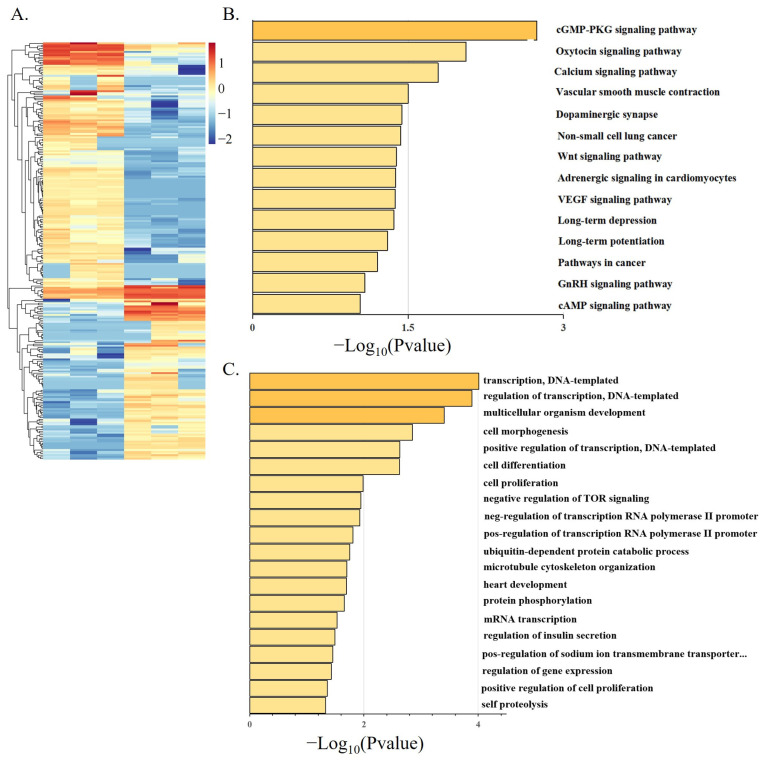
Differential expression analysis of LncRNA during Shitou goose myoblasts differentiation. (**A**) Heat map of DE-lncRNA. The heatmap was constructed using Log10 (FPKM) values.; (**B**) enriched KEGG pathway of DE-lncRNAs; (**C**) GO enrichment of DE-lncRNAs.

**Figure 6 animals-12-02956-f006:**
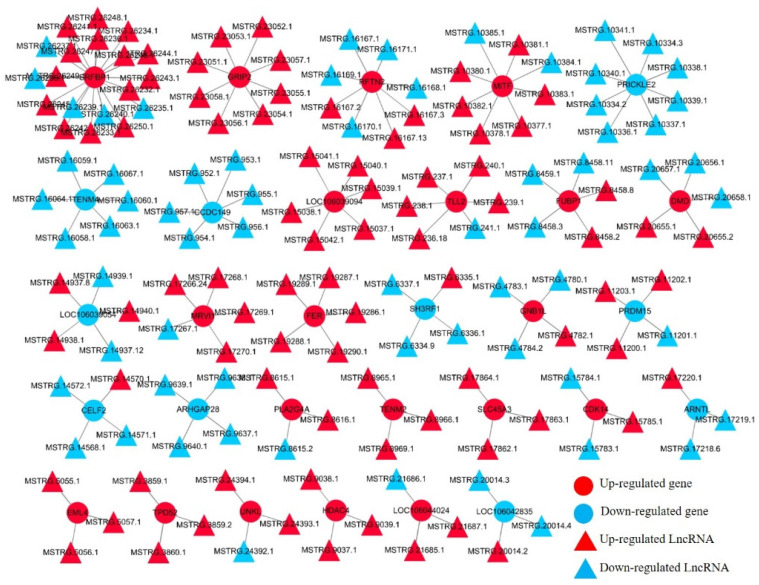
A DE-lncRNA-DEGs interaction network during goose myogenesis.

**Table 1 animals-12-02956-t001:** A comparison of sequencing data among samples.

Samples	Raw Data (Read)	Valid Data (Read)	Mapped Reads	Unique Mapped Reads	Q30%	GC Content %
GM_1	70689352	63945928 (90.46%)	60857652 (86.13%)	57736805 (72.17%)	98.70%	47%
GM_2	79467030	73247104 (92.17%)	69218212 (85.90%)	59177117 (73.44%)	98.62%	48%
GM_3	74221168	68726348 (92.60%)	60857652 (86.13%)	52009319 (73.61%)	98.63%	48%
DM_1	86604440	80002650 (92.38%)	55637470 (87.01%)	47301347 (73.97%)	98.47%	47%
DM_2	89152764	80578354 (90.38%)	63493532 (86.68%)	53952475 (73.66%)	98.57%	47%
DM_3	78730378	70658744 (89.75%)	59425642 (86.47%)	50403679 (73.34%)	98.76%	47%

**Table 2 animals-12-02956-t002:** The top ten DEGs.

The Top Ten Genes Up-Regulated during Myoblast Differentiation			The Top Ten Genes Down-Regulated during Myoblast Differentiation		
Name	Log2FC	−log_10_(*padj* Value)	Name	Log2FC	−log_10_(*padj* Value)
*NMUR2*	5.003212	2.054442	*ESPN*	−5.89484	3.649857
*SLC14A2*	4.787548	3.415658	*MPZL3*	−5.83747	3.542505
*FIGF*	4.586098	59.866965	*SCARF1*	−5.62258	3.159790
*MYRFL*	4.519206	2.034318	*GFAP*	−5.58841	3.122162
*SHANK1*	4.505554	1.395063	*FAM83A*	−5.31	2.557658
*NYAP2*	4.505405	5.634583	*SFN*	−5.28109	3.142785
*GADL1*	4.346948	3.736134	*IRF6*	−5.22921	2.558212
*WNT11*	4.222275	35.025755	*TMPRSS4*	−5.14386	2.382782
*KCNJ16*	4.113374	10.443003	*SDCBP2*	−4.98299	2.099845
*CKMT2*	4.087084	48.253844	*GJA4*	−4.86464	1.923671

**Table 3 animals-12-02956-t003:** The top ten DSGs.

Exon Skipping		Intron Retention		Mutually Exclusive Exon		Alternative 5′ Splice Site		Alternative 3′ Splice Site	
Name	LncLevel Diff.	Name	LncLevel Diff.	Name	LncLevel Diff.	Name	LncLevel Diff.	Name	LncLevel Diff.
*SORBS2*	0.508	*LOC106031985*	0.508	*SORBS2*	0.721	*SORBS2*	0.574	*RBM27*	0.719
*KCTD17*	0.788	*LOC106047490*	0.434	*KLF12*	0.667	*GZF1*	0.551	*PTPN3*	0.667
*AP1S2*	0.755	*MOB3C*	0.425	*LOC106042308*	0.464	*ARPP21*	0.551	*RAPGEF1*	0.603
*REEP1*	0.659	*ZNF644*	0.415	*HENMT1*	0.436	*SYNCRIP*	0.537	*LOC106032270*	0.59
*CAMSAP2*	0.627	*SLC39A3*	0.406	*ACSL6*	0.431	*LOC106031985*	0.465	*MPP3*	0.567
*PBX3*	0.625	*TMEM40*	0.382	*NCAM1*	0.423	*LOC106031985*	0.463	*PRLR*	0.55
*FAM76B*	0.624	*IRF1*	0.376	*ABLIM3*	0.41	*PLAT*	0.448	*CYSLTR2*	0.535
*NCAM1*	0.618	*NFKBIZ*	0.343	*FHL5*	0.365	*SLC39A10*	0.441	*PRLR*	0.529
*TPD52L1*	0.607	*LOC106044164*	0.336	*PCNXL2*	0.36	*LOC106032100*	0.441	*PRLR*	0.523
*EHMT1*	0.606	*SLC6A1*	0.333	*EPHB6*	0.356	*HDAC9*	0.439	*LOC106031985*	0.518

**Table 4 animals-12-02956-t004:** GO analysis of differentially spliced genes.

GO_Term	DS-Gene	Count	FDR (Adjusted *p*-Value)
extracellular matrix	*SMOC2, POSTN, COL26A1, ADAMTS13, COL24A1, ELN, TNC*	7	0.00062
Costamere	*SVIL, CMYA5, DMD*	3	0.01734
actin cytoskeleton	*SVIL, ABLIM1, MYO1C, ABLIM2, DST*	5	0.01997
actin binding	*SVIL, ABLIM1, MYO1C, ABLIM2, DST, DMD*	6	0.03799
extracellular matrix structural constituent	*POSTN, COL24A1, ELN, TNC*	4	0.04167

## Data Availability

The datasets supporting the conclusions of this article are available in GEO: https://www.ncbi.nlm.nih.gov/geo/query/acc.cgi?acc=GSE213147, public on 15 September 2022. The data analysis pipeline (Linux code and R scripts) is provided as a zip file in the Appendix A.

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
