# Peer review of "Transcriptome Sequencing Reveals Pathways Related to Proliferation and Differentiation of Shitou Goose Myoblasts"

_animals, 2022, doi:10.3390/ani12212956_

Round 1

Reviewer 1 Report

Dear Authors,

thank you for possibility of reading your work!!!

First Figure 1 and 2 are blurry and very hard to read from them. Especially Figure 2 it must be divided because those informations are totally unable to read. I know that this may be hard as there are many results presented but You must find some solution - maybe A and B put into Figure 2, C and D into Figure 3. E as Figure 4, F as Figure 5 and G and H as Figure 6.

Also Figure 5 must be somehow changed -  i don't see names next to circles adn triangle

Also in M&M part - you describe Isolation and culture of myoblasts - I'm not very friendly with cell culture but I miss some informations here - how may embryos you used? Every embrio was a cultured on one culture dish?.

Also in part 2.2 you wrote that you extracted RNA from goose tissues and cells - there were some other samples than cultured myoblasts? Also qPCR were made to confirm results from Illumina?

2.6 -"Each sample was analyzed based on results that were repeated at least three times" - how many samples were analysed and what kind of samples? also I understand that every samples were sequenced three times? And statistically significances - why three levels (0.05, 0.01 and 0.001)? 

Also in supplementary information 1 - when I opened this file I found some date (day/month)

best regards

Author Response

Thank you for the comments.

Based on your comments, we have carefully revised our manuscript (animals-1885561). The itemized responses to each comment are addressed one by one as follows. Many thanks for your suggestions.

Best wishes,

Wen Luo  

Author Response

(The authors gave the same response as above.)

Round 2

Reviewer 1 Report

Dear Authors

Thank you for your manuscript revision. For me it is sufficient to be publish 

Author Response

Thank you for the comments.

Best wishes,

Wen Luo 

Reviewer 2 Report

Dear Luo,

In second round of review, I could eventually see the figures in a readable resolution to complement  first round of review and thus I could provide more comments/questions about your study. According to policies at Animals, you needed to provide point-by-point responses to reviewers comments. I could see some of them have been addressed in the main text. But some comments have not been addressed. Hence, I would like to invite you to go through both reviews and respond. 

Regards,

Author Response

(The authors gave the same response as above.)

Round 3

Reviewer 2 Report

Dear authors,

The paper has improved to large extent however I still have concerns which I have expressed in attached file where you can find some errors too. 

Regards,

Author Response

(The authors gave the same response as above.)
